# Emerging from the COVID-19 Pandemic: Aviation Recovery, Challenges and Opportunities

**Kaitano Dube**

Ecotourism Management, Faculty of Human Science, Vaal University of Technology, Vanderbijlpark 1911, South Africa; kaitanod@vut.ac.za; Tel.: +27-710096290

**Abstract:** This exploratory study examined the impacts of COVID-19 and emerging challenges and opportunities from aviation recovery. Using archival and secondary data analysis, the study found that there are several challenges to aviation recovery chief among them are labour challenges and extreme weather events, which have been responsible for traffic disruptions in major aviation markets such as Europe and the USA. Other emerging challenges include high debt, inflation, interest rates, fuel, cost of labour, and general operational costs. The study recommends several interventions to address the sector's challenges, including adopting risk disaster preparedness and management to foster sustainability.

**Keywords:** labour challenges; climate change; Russia–Ukraine war; high fuel costs; sustainability

## 1. Introduction

COVID-19 had a devastating impact on the travel and tourism industry across the world. Various destinations, sectors and subsectors of the travel and tourism industry were adversely affected by the pandemic with a varying magnitude [1]. At the pandemic's peak, various governments and states responded differently to the pandemic to ensure survival [2]. Even though the initial shock of the pandemic could have been uniform across various economies due to the hard lockdown approach adopted by various countries [3]. Post the initial shock, the experiences of living with COVID-19, in many respects, have been unique for each country and region, given differing vulnerabilities and resilience.

Besides affecting the economies, the COVID-19 pandemic also altered tourists' travel patterns and behaviours [4,5]. This further complicated situations for many tourism enterprises globally. COVID-19, apart from triggering a wave of lockdowns [6], also triggered phobia for travel amongst regular travellers, and many vulnerable population groups were discouraged to travel particularly long distances [7]. According to Bin et al. [8], COVID-19 also triggered a culture of virtual participation and meetings, adversely affecting the travel industry. Virtual events ballooned and grew during the peak pandemic, primarily driven by the internet. This inadvertently has long-lasting implications for travel trends, most in an adverse manner. Almlöf et al. [9] argued that, in the main, the pandemic forced many to abandon public transport usage to avoid crowded places. In other settings, the pandemic promoted greater usage of alternative transport, such as private vehicles and bicycles [10]. There is no doubt that this adversely affected public transport service providers who had to deal with reduced number of consumers with potential adverse impacts on operational viability and profitability.

The advent of COVID-19 vaccinations offered hope for many travellers as it boosted travel intentions in many countries worldwide [11]. The advent of the vaccine in late 2020 and early 2021 and the rollout of COVID-19 health and safety protocols coincided with a decline in COVID-19 infections. This gave hope for a new start and recovery for several economic sectors, including the travel and tourism industry. According to Ekinci et al. [12] and [7], the global economic recovery hinges largely on attaining herd immunity. As the

vaccine rollout ensued, it became clear that the recovery, if ever it were to happen, was going to be complex given the staggered approach to which most destinations opened and other effects of the pandemic. This hugely affected travel patterns, particularly in the aviation industry, which depends on well-connected networks globally.

The pandemic reshaped airlines' route networks and aircraft [13], which woke up to a new reality or the so-called new normal. As feared and postulated in the early days of the pandemic, the disruption caused by COVID-19 brought new chaos and order, which aviation and other tourism companies had to grapple with. This study seeks to examine the impacts, recovery challenges and opportunities faced by the aviation industry as it grapples with coming out victorious from the impact of COVID-19. The study is critical as it traces the journey travelled and offers insights into the future trajectory of this critical economic driver.

## 2. Literature Review

The aviation industry plays a central role in the global economy [14,15] as it acts as a trade, industry, and collaboration vehicle. The pre-COVID-19 tourism growth witnessed before 2020 was largely driven by the growth of the aviation industry and air connectivity [16,17]. The aviation value chain has positive economic spin-offs on other facets of the economy. As such, robust aviation industry is beneficial by and large to the economic prospects of a country or region [18]. This makes the welfare and well-being of the aviation industry a global concern.

The COVID-19 pandemic had a debilitating impact on the global aviation industry in the main. The impact of COVID-19 by far surpassed previous experiences with other aviation shocks, such as the SARS [19]. The pandemic resulted in declines in air mobility in many parts of the world for both civilian and military flights (ibid). According to [20], the COVID-19 pandemic's impact on aviation has been much more pronounced since it disrupted the global supply chain. The disappearance of airlines also adversely impacted the collection of weather data in some regions of the world [21]. This could pose challenges for the aviation industry which depends on weather and climate data in flight path planning and also create climate data gaps at a time when there is a need to understand climate patterns to plan for climate change adaptation.

Despite the negatives of the COVID-19 pandemic in some respects, the pandemic resulted in unintended consequences. There is new evidence that suggests that in as much as there are several airlines that went bankrupt as a result of the pandemic. The disease outbreak caused the mushrooming of new airlines in some parts of the world. Sun et al. [22] noted that the pandemic ushered in a whole new era of airline start-ups which came largely at the expense of old airlines that had a better risk appetite and were innovative in their operations. Several of the new start-ups seem to have been in Europe and Asia. These present policy challenges for the sector in many respects (ibid).

There is a need to acknowledge that the pandemic, apart from unleashing terror on the tourist market, also created a whole new business culture and ecosystem. It forced a rethink of the tourism and aviation business model as we know it and resulted in many questioning the sustainability of the aviation business model. At the height of the pandemic, many employees in the tourism and aviation industries were laid off through retrenchments [23,24]. Many of the laid-off staff were skilled and experienced. Indeed, other staff members succumbed to the disease, while others opted for other jobs altogether. Many academics warned that the post-pandemic tourism aviation industry was likely to be transformed [25] and, in many ways, a departure from the old traditional industry.

Given that COVID-19 altered many aspects of people's lives, including those working in aviation tourism, there is a huge demand to understand how aviation companies were fairing during recovery. There is also a need to understand some of the innovations, successes, and challenges of the new normal, which is technology driven. Such learnings are key for policy, practical interventions, and future pandemic lessons. It is against this background that this study is conceptualised. With the new normal after some severe cash

burn, new challenges emerged, and so are the new opportunities that this study seeks to explore. The study emerges from the highly touted uncertainties that were highlighted post the emergency of the COVID-19 pandemic [22].

## 3. Materials and Methods

The study makes use of archival data. The aviation industry produces many data on a minute-to-minute basis as data is generated from online search engines, bookings, ticket sales and mobility activities. Therefore, there is ample data that can be analysed and used for policy and practice within tourism and aviation that is instantaneously produced daily. From an aviation perspective, the aircraft's landing, departure and movement produce adequate data that can be used to make a timely decision based on the availability of this data to the right user. This makes it inexpensive for researchers to utilise this data in the industry and for academic studies.

This study utilises data from authoritative sources which collect this information from industry players such as airlines, air navigation companies and airports, amongst other such sources. Therefore, the aviation industry produces a lot of big data available for usage by various stakeholders. A full-blown primary data collection for this research would also be almost impossible, given the sheer size of the study area. It would not necessarily yield any different results than those presented here.

This study utilises data from archives and reports from EUROCONTROL and International Air Transport Association (IATA). (ATA), was founded in 1936. On the other hand, EUROCONTROL is an aviation intergovernmental organisation comprising 41 Member States and two states with observer status. It provides air navigation services to military and commercial civilian airlines that operate within the Eurozone. The organisation manages the European ATM Network (with nearly ten million flights annually) in close liaison with air navigation service providers, airspace users, the military and airports [26].

The study also used additional data sources from aviation blogs and international aviation data sources such as IATA, Flightradar24, FlightAware and other key aviation sources such as Notice to Airmen (NOTAM). The International Air Transport Association (IATA) is a trade association of the global airline industry that was set up in 1945.

On the other hand, Flightradar24 is a Swedish-based company that tracks global flights. It has the largest ADS-B network and utilises 35,000 connected receivers. Consequently, aircraft manufacturers such as Embraer, Airbus and Boeing utilise their services. It monitors about 200,000 flights daily [27]. FlightAware is an American technology company which provides accurate real-time flight tracking data products to all sorts of aviation role players. It also provides modelled flight and historical data. It utilises 32,000 ADS-B ground stations in 200 countries [28]. These data sources account for a large portion of the global aviation industry and, as such, can be considered to be broad enough to offer insight into what is happening within the aviation industry. The bulk of the data that was analysed was from 2019 to August 2022.

Content and thematic analysis was the principal analytical tool used to make meaning out of the data. The study used research questions and sub-research questions to acquire relevant data for this study. Additional analysis was done using Microsoft Excel Toolpak. Of interest was to explore the data that talks directly to the impacts and challenges remedies adopted by the aviation industry in response to the impacts of the COVID-19 pandemic.

## 4. Results

The study found that the pandemic's worst impact was felt in 2020, with recovery starting the same year in the second half of the year (Figure 1) and continuing in 2021 and peaking up pace in 2022. Evidence shows that significant recovery progress has been witnessed, with the aviation industry edging closer to the levels that were witnessed in 2019, which act as a baseline for the aviation industry on which the impact of COVID-19 is based. The prolonged decline in volumes of traffic recovery teethers means that aircraft were parked and not making as much revenue. This adversely affected the aviation industry's

value chain, adversely posing challenges for recovery. The long period of cash burns resulted in some airlines, airport companies, businesses and employees of various aviation entities facing sustainability challenges. This is confirmed by previous studies, which state that the impact of COVID-19 on aviation was adverse across the value chain [13,29,30].

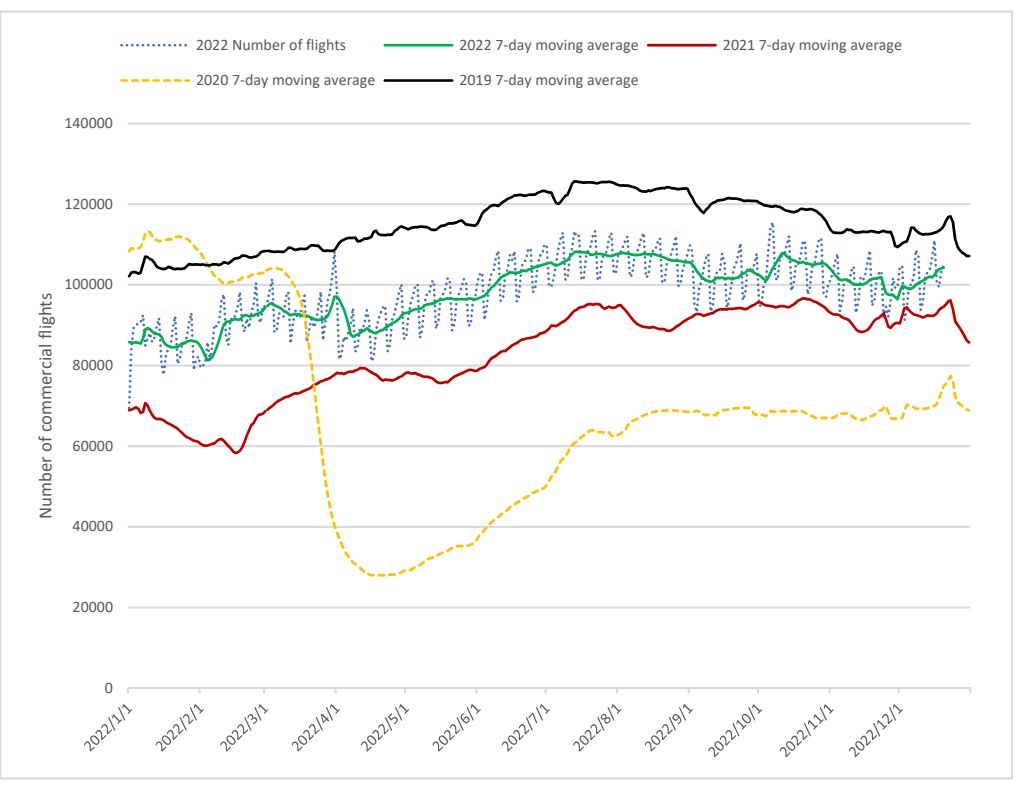

**Figure 1.** Global Traffic evolution impact and recovery from COVID-19 Source: Author Data from Flightradar24.

The COVID-19 pandemic reset the aviation and tourism industry in many respects and the impacts varied across global regions, which created various challenges, some unique and some generic to the sector as a whole. The following sections look at the evolution of air traffic in some of the biggest global aviation markets.

Evidence shows that global international travel started recovering after June/July 2020. Significant positive gains were also witnessed around November 2020 (Figure 2). However, there is a slump soon after that due to the impacts of the Delta variant that was 1st detected in India in late 2020. Significant growth was witnessed thereinafter June of 2021, which has been largely sustained in many aviation markets except in the Asian market, which has sustained a rather slow growth reaching—60% with several other markets reaching between −20% and −10% (for Africa, Europe, Middle East, North America and Latin America and the Caribbean). The long recovery period in the Asian market is a factor of China's Zero COVID-19 policy which proved costly for the travel and tourism industry. Evidence suggests that the passenger volume for domestic travel has been more robust as compared to the international tourism market. Between 21 January and 21 May, Europe's seat capacity reduction was much lower than the rest of the world (except Asia/Pacific). Domestic travel has thus far recovered to around +/−20%, with other regions surpassing the pre-pandemic levels. Latin America and Europe's domestic travel seem to have outpaced the pace of travel before 2019. This could be attributed to the fact that most travellers were forced to embark on domestic journeys to evade international travel, which came with several conditions and fears and anxieties. Nonetheless, the recovery has been punctuated by declines, probably mirroring restrictions that various governments imposed to curtail the spread of the pandemic. On the other hand, the market tended to respond adversely to

spikes in infections often characterised by discovering new COVID-19 variants, which meant stricter control measures.

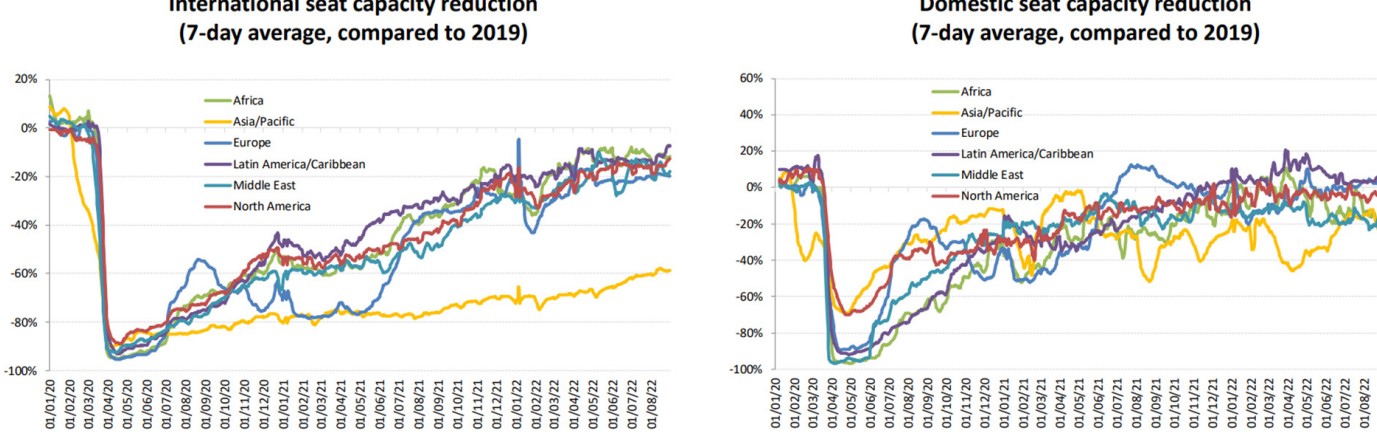

**Figure 2.** Seven-Day Change (%) vs. 2019 in Onboard Passengers Source: ICAO.

In Mexico, however, the recovery surpassed the 2019 levels in the main, with passengers onboard airlines mainly in the positive territory. The domestic travel market has significantly recovered, with peaks in mid-year 2020, April 2021 and March 2022 after the Omicron variant's discovery in late 2021 and early 2022. There is evidence that the domestic USA tourism market has edged closer to 2019 levels showing evidence that there is some steady confidence in domestic travel.

This market development is not unique to the aviation industry, but other sectors have experienced the same trajectory where domestic tourism is the first to recover. This confirms the earlier anticipation that COVID-19 would trigger a different trajectory regarding tourism recovery. Woyo [31] argued that post-COVID-19, domestic travel offered one of the best options for tourism sustainability. The industry and uncertainty imposed by the pandemic also pushed several people into opting for more localised travel to avoid the complications of international travel amidst the raging pandemic. This kind of tourist behaviour has implications for the type of aircraft that airlines can use. Evidence from Flight Radar and FlightAware indicate that most airlines opted to deploy their much more fuel-efficient and medium-sized aircraft.

In the European market, the cargo was not as severely adversely impacted by the pandemic as other segments of the sector; the overall picture shows that the sector benefitted from the pandemic as it witnessed growth in business despite the pandemic (Figure 3). Cargo operations remained largely in the positive territory throughout the pandemic. The other segments of aviation experienced the initial shock caused by the pandemic, which resulted in a steep decline in March 2020. Nonetheless, the recovery trajectory has been different across the various segments of the aviation sector. Business aviation recovery started in April, soon after the sharp decline in March 2020. In July 2020, the segment surpassed the 2019 figures, although it declined soon after that, peaking up into positive territories during the Christmas period in December and January in 2020 and 2021. As of June 2021, business aviation has been operating way above the pre-COVID-19 levels. This demonstrates the resilience of that particular market. Business travel also offers more secure travel as it limits the interaction with the majority, for example, in the conventional airline industry, with the addressing of traveller fears and the demands for crowd avoidance for health safety reasons.

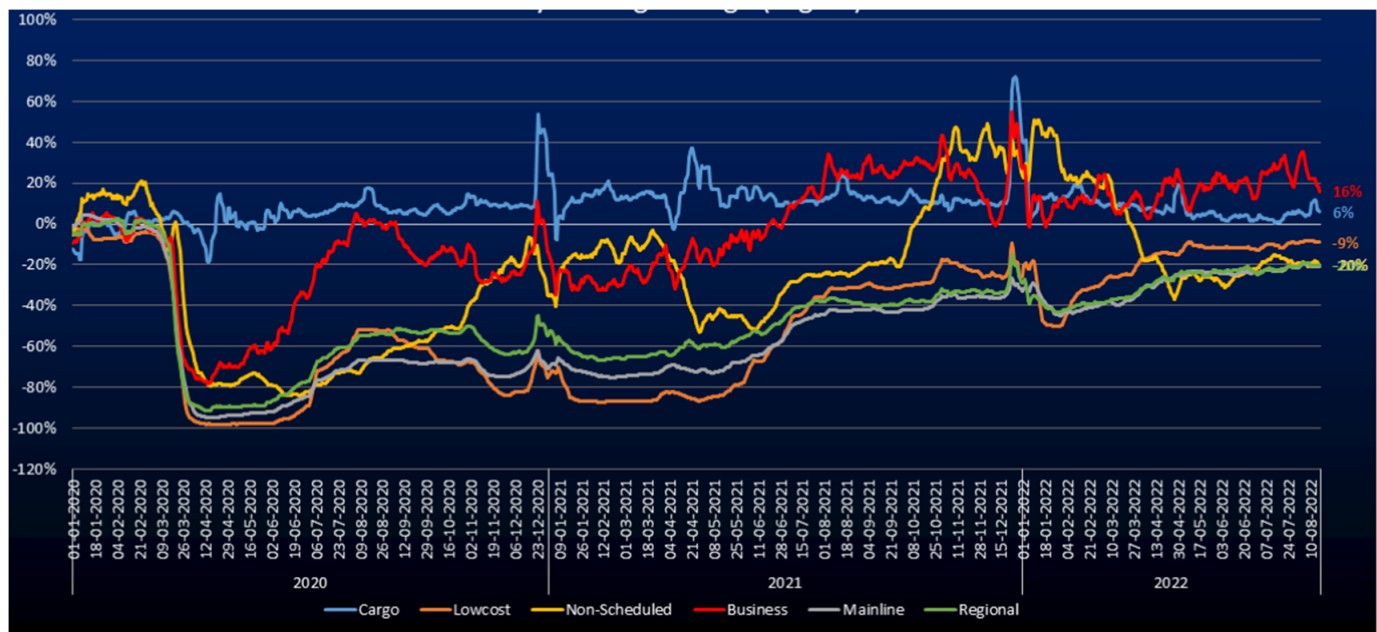

**Figure 3.** EUROCONTROL Market performance in percentage and COVID-19 recovery Source: EUROCONTROL (2022:6).

Charter flights also seem to have witnessed better recovery. However, there are signs that this sector was sensitive to variations of the pandemic, probably because of the consequent travel restrictions, which limited mobility. Between October 2021 and March 2022, the sector was operating better than in the pre-COVID-19 situation. The downward spiral seems to have been triggered by the Russia–Ukraine war, which started on 20 February 2022. The war triggered global economic turmoil, which resulted in global market pressure characterised by high inflation and fuel costs and subsequent interest rate hikes.

There has been generally a slow recovery regarding other aviation segments, including low-cost, mainline, and regional airlines. There is evidence of adverse impacts of different variants, including Omicron, which had a pushback effect on the recovery process in late 2021 and early 2022. However, the general picture shows a general recovery pattern in the sector's trajectory, which points to improvements. As of 8 August 2022, low-cost airlines had recovered better and trailed behind the 2019 levels by −9%, while regional was −21% and mainlines trailed at −20%. The uptake in low-cost airlines could be attributed to more people utilising them, given the affordability challenges induced by COVID-19. Given the inflationary pressures associated with the COVID-19 recovery period, consumers were looking for cheaper deals to offset the financial squeeze. Mainline and regional airlines remained challenged by changes in travel behaviour, consumer confidence and restrictions, which hampered wholesale travel by tourists.

*4.1. Travel Recovery Challenges*

Aviation recovery came with many challenges, some of which were not anticipated at the onset and before the pandemic. One of the biggest challenges the aviation industry had to battle was the increased cost of doing business induced by the pandemic and other external factors directly and indirectly attributed to the pandemic. When the pandemic started, several airlines and aviation companies did not have adequate financial resources to fund operations during the international shutdown in March 2020. Airlines had to resort to other mechanisms to fund unavoidable cash burn. Some airlines were forced into adopting measures such as furloughing and retiring staff and older aircraft to better manage their cash reserves and debt associated with aviation operations at that time [32,33].

COVID-19 related disruptions in the aviation and tourism industry led many to question the sustainability of working in the tourism and aviation sector given the lack of safety nets and job security [34,35]. As consequence at the height of the pandemic, many employees were retrenched or furloughed. This led to disgruntlement and many furloughed employees looking for jobs in other economic sectors with better working conditions and job security. As a consequence, the industry lost many experienced and knowledgeable employees which adversely affected the tourism and aviation recovery process. On the other hand, employees who have been working under difficult conditions in the tourism and aviation industry characterized by poor salaries and poor working conditions were demanding better wages and working conditions. This resulted in industrial action in the form of job strikes and protests at several airlines and airports in 2021 and 2022. This resulted in traffic disruptions which inconvenienced many travellers. The shortage of skilled personnel also adversely affected airlines and airports in the USA and Europe. Heathrow Airport International, for example, was forced to put a cap on the number of travellers per day, limited to 100,000 per day [36]. The capacity limitations were put in place from July 2022 and were set to be in place until October 2022.

The capacity cap was placed to deal with staffing challenges as the passenger volumes grew faster than anticipated. This resulted in chaos at the airport, which resulted in luggage losses and delays and some luggage not travelling with their owners. Other challenges were long queues at check-ins and baggage collection for arriving customers. Other airports that placed capacity caps to deal with a huge influx of people moving to catch up with the backlog imposed by COVID-19 include Gatwick, Frankfurt and Schiphol in Europe (See Figure 4). In mid-August 2022, it was reported that there were about 21,000 flight delays and 1600 cancellations [37]. In the USA, there were 20% flight departure delays at four airports, and the highest flight delay was 36% at Denver International Airport (Ibid). Such disruptions placed many inconveniences on the travelling public, with huge financial losses for all the tourism stakeholders.

Figure 4 shows that apart from staffing and capacity challenges which accounted for the majority of delays accounting for about 47% of the delays and other delays which could be attributed to operational related delays such as technical issues there seem to be evidence of growing challenges of weather-associated disruptions at airports across Europe. EUROCONTROL blames 23% of the weather-related departures on Cumulonimbus (Cb). The heating of air close to the surface is one of the prerequisites for Cb formation. Europe has witnessed record-breaking temperatures between June and July 2022, with many stations recording record temperatures, some as high as 40 °C. A study by Zachariah et al. [38] observed that climate change made the temperatures in Europe in the year 2022 10 times more likely, and without climate change, temperatures observed in Europe would have been 2 °C less hot. The heat in Europe coincides with increased Cb weather disturbances (Figure 4). Given the relationship between heating and Cb development, it can be argued that climate change is a factor in the observed weather-related disruptions to aviation in Europe during June and July of 2022. Evidence shows that there was an increase in weather-related delays in 2022 in comparison to 2021 which was cooler than 2022 [19,39]. The persistent challenge of Cb is a challenge for aviation as it triggers air turbulence, can cause electrical challenges for aircraft and cause inflight icing. Given the harm caused by the same, pilots often resort to rerouting in an avoidance tact, resulting in increased fuel burn and the associated costs.

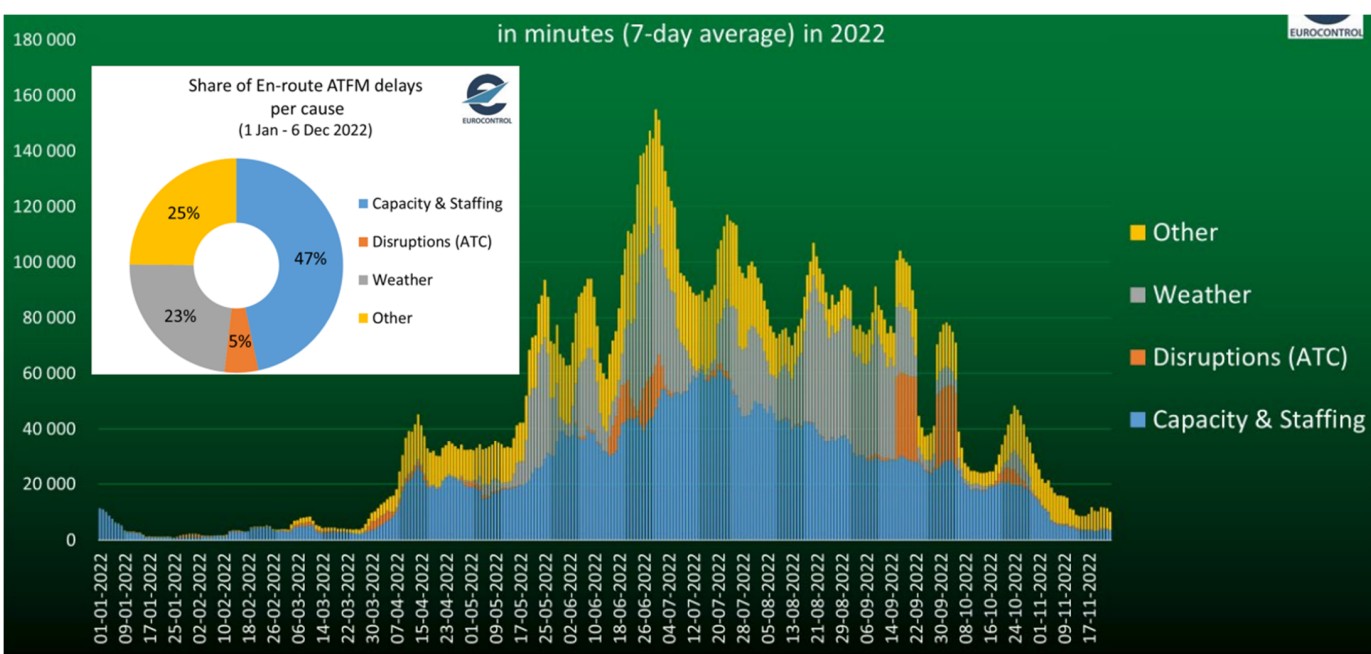

**Figure 4.** Europe Enroute ATFM Delays in minutes 7 days average Source: Reprinted with permission from EUROCONTROL [39].

There is also an increase in flight delays from other reasons such as Air traffic control (ATC) and other disruptions. It is unclear whether these are related to labour or other COVID-19-related challenges. In many respects, the recovery period came amid growing global inflationary pressures and high-interest rates (Figure 5), raising the cost of living. In order to attract back employees and new skills, the aviation industry had to push up salaries and wages to make it attractive for personnel to work in the aviation industry. The struggling airlines had to battle high wages bills, which hampered the recovery process

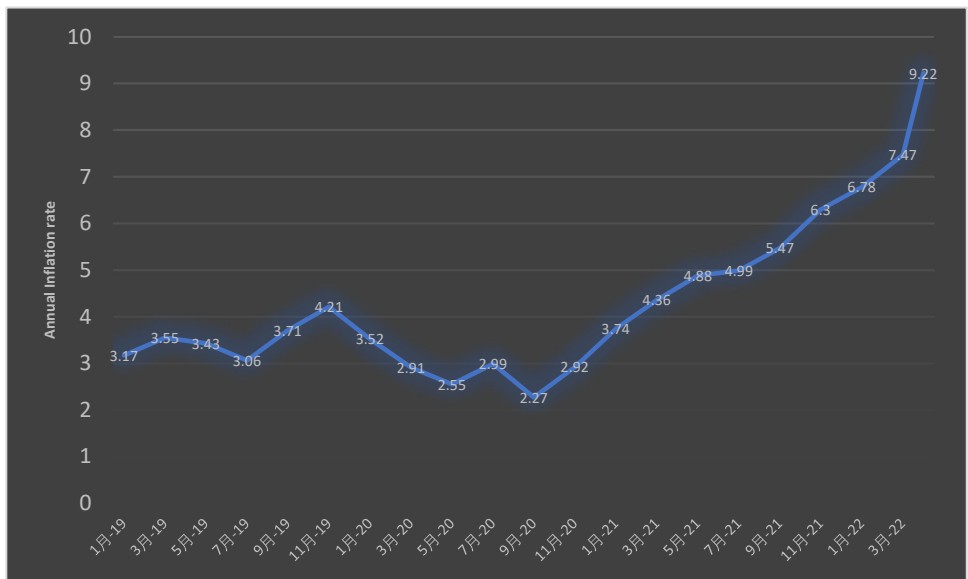

**Figure 5.** Global average inflation rate: Source: Authors Data from Statista.

The hyperinflation environment came with another challenge not seen during the height of the pandemic season. At the height of the pandemic season, global oil prices tanked, providing some relief and allowing many airlines to remain afloat [20]. Due to geopolitical factors, oil prices rose significantly in the first quarter of 2021, which offset the

little reprieve that had taken root at the height of the COVID-19 pandemic. The tension and the ultimate outbreak of the Russia–Ukraine war and subsequent efforts to punish Russia by the Western countries and the USA further complicated the fuel prices and led to even more fuel price increases in early 2022 (Figure 6). Generally, the trend of the price of jet fuel shoots up significantly during the time aviation was trying to recover from the adverse impacts of COVID-19. This subsequently increased operational costs for airlines. This threatened the survival and sustainability of airlines, particularly those that did not have access to financing. The sharp increases in fuel came at a time when airlines were making frantic efforts to service debts accrued at the height of the pandemic.

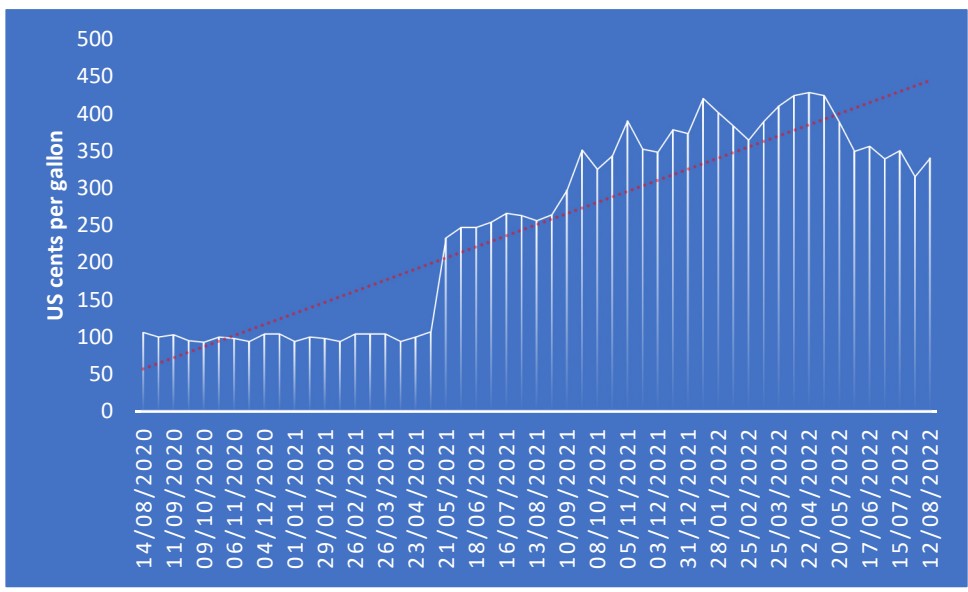

**Figure 6.** Jet Fuel Price 2020 to August 2022 Source Authors Data from IATA.

In a bid to tame inflation, central banks across the world responded by increasing interest rates. This development had adverse impacts on debt repayment by aviation companies as it raised the level of debt by these companies. The increased interest rate is a challenge to consumers whose purchasing power has been equally eroded, which could adversely alter travel intentions. This will worsen the already suffering but recovering aviation sector, which is only expected to return to profitability in 2023 and is led by America, which is set to record profits in 2022. According to IATA [40], North America is set to witness profits of $8.8 billion in 2022. Other regions are expected to witness losses, with Europe expected to record losses of $3.9 billion, Asia-Pacific at $8.9 billion, Latin America at $3.2 billion, the Middle East at $4.7 billion, and Africa expected to witness losses of $0.7 billion. Profits in American aviation are predicated on the robust tourism market serviced by their airlines and the approach the government took in the management of COVID-19.

*4.2. Opportunities Presented by COVID-19 in the Aviation Context*

COVID-19 transformed several industries with many adverse impacts, which will take some time to rectify. Undoubtedly, these challenges came with opportunities that can advance industry aspirations. One of the accusations which have been levelled against the aviation industry is that it is carbon intensive and significantly contributes to climate change, and in many respects, unsustainable in many regards. The decline in aviation traffic came along with significant reductions in carbon emissions. In many respects, there is a need to ensure some areas have significantly lower carbon emissions than in 2019 (Figure 7), which is the base year. IATA estimates that carbon emissions from global aviation will rise to 809 million tonnes, up from its declines in 2020 and 2021, when $CO_2$ fell to 495 and 577 million tonnes, respectively. In 2019 the aviation industry had total carbon emissions of 905 million tonnes [41]. The quest for NetZero is one of the global aspirations that every

aviation manager is clamouring for to meet investor and client demands for sustainability within that sector.

**Figure 7.** $CO_2$ variation from January to June 2022 vs. 2019 departures. Source Reprinted with permission from EUROCONTROL [39].

The industry can leverage some retired old aircraft to purchase and bring on newer, more fuel-efficient airlines. It is therefore encouraging to note that of the aircraft delivered in 2021, the bulk of them was in the high fuel efficiency category. Airbus, for example, delivered 459 aircraft from its neo fleet range and 221A321ceo while Boeing delivered 245 of 737 MAX [42], which are considered some of the most fuel-efficient fleets on the market. This will help airlines to move towards NetZero.

Riding on the demand for sustainability consciousness that has been reignited by the COVID-19 pandemic [43], the aviation industry can use the momentum to lobby for greater environmental actions concerning cutting carbon emissions. Tourists can be encouraged to take initiatives such as offsetting their carbon emissions, mainly through market-based measures and purchasing carbon credits. Greater efforts can also be made to lobby tourists to as much as possible travel light with regard to travelling light as part of the new culture of travel that is environmentally sound. Such efforts can assist in reducing the environmental cost of the aviation sector. Most importantly such an approach will also unlock funds for climate change mitigation. Such funds can be crucial in developing countries in particular which have huge funding gaps for both climate change mitigation and adaptation.

There is a lot more than the aviation industry can do to leverage the current situation. The sector should use the time to invest in efficient route operations in preparation for the full sector's recovery. This will save airlines money and also assist in reducing flying's carbon footprint. Research and innovations in Sustainable Aviation Fuels can be ramped up with a view of ensuring that the costs for the same drastically come down to near prices of jet fuel or avgas as current prices are prohibitive. Rolling out distribution infrastructure for the same will assist a great deal in nudging the sector towards sustainability.

The COVID-19 pandemic also ushered in the heavy deployment of technology within the aviation sector [44]. This includes using robots, facial recognition technology and other biometric technology to ensure contactless travel. Other technologies rolled out

include retrofits and advances in aircraft engines to improve efficiency and reduce carbon emissions as part of the sector's sustainable recovery approach. The industry can continue to progress and promote smart travel at an increased pace which can assist in better travel and ensure the industry is well-prepared for future pandemic events. Various advantages also have been witnessed through the COVID-19-induced technology system, which increases security and assists aviation in moving towards a paperless sector. Moving away from paper can also help the industry in reducing its carbon footprint.

Aviation can also leverage lessons from the current pandemic to foster greater flexibility in terms of tickets and booking and provide a better customer experience in the manner most airlines accommodated passengers at the height of the pandemic. Such a move can be employed by airlines to address challenges imposed by extreme weather events whose impact on the aviation industry has been on the increase in recent years [45–48]. The current labour challenges for the sector can be used to foster better working and wage conditions for the aviation industry and pace the rate at which the industry addresses the sustainability challenges it faces [49] in line with the dictates of the Sustainable Development Goals agenda. Part of the challenges that the action faced at the height of the pandemic was severe liquidity challenges [50,51]. Airlines, airport companies and other stakeholders can take lessons and better prepare and push for increased cash reserves which will allow the airline to build better resilience should there be such catastrophic events in the future.

## 5. Conclusions

The study sought to assess the challenges and opportunities faced by the aviation industry to recover from the COVID-19 impacts. The study found that the impact and recovery of COVID-19 have not been uniform across the world. In some destinations, the recovery was slow owing to various factors amongst differentiated control measures aimed at containing the spread of the COVID-19 pandemic. Global challenges are facing the aviation industry as it seeks to recover. Chief amongst them is geo global political and economic factors such as increased interest rates, inflation, the rising cost of fuels, labour challenges and the Russia–Ukraine war, among other factors. These factors, including climate change, complicated the recovery process, threatened to slow the recovery process, and could throw some aviation players out of business as continued disruption of the sector threatens its short-term viability in some geographic regions.

The study identified several areas where the aviation industry could take advantage of the situation of COVID-19 when it disrupted the sector. Amongst some of these areas is the need for the industry to increase the pace of technological innovation, and there are opportunities to address climate change and other sustainability issues. Given lessons learnt from the pandemic, there is a need to deal with increasing the financial buffer reserves to allow the sector to respond to future disasters and the pandemic. The aviation industry needs to think and put better safety nets to improve resilience in existential threats such as pandemics and other natural disasters.

**Funding:** This research received no external funding.

**Data Availability Statement:** Data available on request.

**Conflicts of Interest:** The authors declare no conflict of interest.

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
