# Peer review of "Emerging from the COVID-19 Pandemic: Aviation Recovery, Challenges and Opportunities"

_aerospace, doi:10.3390/aerospace10010019_

Round 1

Reviewer 1 Report

This is good paper that fits the theme of the 4th International Aviation Management Conference. The topic of the study is contemporary and there some elements that can provide insight into how the industry should better deal with future crises.

There are some improvements which would strengthen the paper as follows:

·        Line 40/41 – The author states that the use of private vehicles and bicycles adversely affected public transport service providers. Please provide a reference to support this statement.

·        Line 73/74 – This is a good point about reduced weather data but could be expanded to explain the impact of this for aviation and the wider society.

·        Lines 132 to 134 should be placed at the end of line 121 to keep information about data from A4A together.

·        Line 158 – Typographical error: ‘…traffic s recovery…’

·        Line 161 – Typographical error: ‘…burn s the aviation…’

·        Figure 1 – It would be useful to include an updated version of this graph that shows the data up to the end of 2022.

·        Line 181 – Spelling error: ‘COCID-19’ should be ‘COVID-19’.

·        Figure 2 (left-hand chart) – The chart shows that between Jan-21 and May-21, Europe’s seat capacity reduction was much lower than the rest of the world (except Asia/Pacific). It would be useful to include an explanation in the text.

·        Figure 4 – The chart is quite blurry and difficult to read and interpret.

·        Figure 5 – Horizontal axis includes an unknown symbol that needs to be corrected.

·        Line 328 to 332 – What are reasons that North America will register a profit while the rest of the world will register a loss in 2022? What is different about the North American market?

·        In Section 3, the use of data from Airlines for America is discussed but there is very little evidence in the rest of the paper about the use of this information source. It would be useful for the author to clarify what data from A4A was used in the study.

·        It is good to see that the author has stated some of the positive impacts of the pandemic on the aviation industry which adds a nice balanced argument to the paper.

·        Figure 1 – Horizontal axis is a little confusing since it includes the year. Instead, labelling the axis in terms of months from January to December would make it clear. Also, it should be made clear which types of aviation traffic are included in the ‘traffic volume’. For example, does the traffic volume include military flights, helicopters, drones, private jets etc?

·        Figure 2,3,4 – Vertical axis labels are missing.

·        Section 5 – It would be useful to include a more detailed discussion on the insights derived from the study which would help the aviation industry to better prepare and manage a future crisis.

Author Response

Dear Sir/Madam

Thank you for making time to review our work. Your efforts to improve the quality of the submission are highly appreciated. Attached is the manner we responded to each of the constructive comments 

Reviewer 2 Report

Overall, this was an interesting and enjoyable read.  My only substantive concerns were regarding the climate arguments that require more evidence to support the author's claims.  This portion (one paragraph) could also be omitted without detracting significantly from the paper.  

Line 166, Figure 1.  The abscissa is labeled incorrectly. I believe it should be month only, not month and year since each line on the chart represents its own year. 

Line 232:   Should be "pre-COVID" not "pe-COVID"

Line 262-263:  The line is confusing as written. Recommend rewriting for clarity. 

Line 280-294:  (MAJOR) This paragraph is speculative and unsubstantiated with data.  In line 282-283 the author states EUROCONTROL blames most the weather delays on Cb for 2022.  More detail should be provided, for example, what is meant by "most" and what are the other categories of weather delays.  The chart provided provides no such fidelity.  In line 289 the author states "the heat coincides with increased Cb weather disturbances." Climatologically, this is when one would expect the greatest thunderstorm activity for any year.  There is no evidence to support that the frequency of Cb during that period in 2022 is greater or less than the same period in previous years.  Thus, the statement in lines 289-290 stating that "climate change cannot be ruled out for causing weather-related disruptions" is weak.  This statement could be made regardless--the data provided doesn't support or oppose the statement.  There is also no evidence to support that the comparatively warmer temperatures in June-July 2022 directly resulted in greater convective activity compared to previous years.  This makes the ties to climate change less compelling. I'm not suggesting climate change is not a factor, the author just needs to provide more evidence to support this claim or omit the argument from the paper.  

Lines 280-294: Recommend replacing "Cumulonimbus (Cb)" with either "thunderstorm activity" or "convective activity."  Cb refers to a single storm or cell, whereas "convective activity" could apply to single cell, multi-cell, or organized thunderstorms (squall lines, MCCs) and their associated impacts (gust fronts, derechos, etc.).  

Figure 4.  This figure needs to be provided in higher resolution.  The pie chart, especially, is not legible.  Also, there is no mention of what "other" includes on the chart. 

Figure 5.  There appears to be an unnecessary character in the abscissa, separating the month from the year. 

Figure 6. The red line should be defined and explained.  The line appears to be a trend line, but it is never discussed.  Also, verify the legend correct?  Is it really cents per gallon?  This makes the value near $1 per gallon in 2020.  Is this crude oil or aviation gas?  Please specify for clarity. 

Author Response

Dear Sir/Madam

Thank you for making time to review our work. Your efforts to improve the quality of the submission are highly appreciated. Attached is the manner we responded to each of the constructive comments.
